# WHICH MODEL TO TRANSFER?
# FINDING THE NEEDLE IN THE GROWING HAYSTACK

## ABSTRACT

Transfer learning has been recently popularized as a data-efficient alternative to training models from scratch, in particular in vision and NLP where it provides a remarkably solid baseline. The emergence of rich model repositories, such as TensorFlow Hub, enables the practitioners and researchers to unleash the potential of these models across a wide range of downstream tasks. As these repositories keep growing exponentially, efficiently selecting a good model for the task at hand becomes paramount. We provide a formalization of this problem through a familiar notion of *regret* and introduce the predominant strategies, namely *task-agnostic* (e.g. picking the highest scoring ImageNet model) and *task-aware* search strategies (such as *linear* or *kNN* evaluation). We conduct a large-scale empirical study and show that both task-agnostic and task-aware methods can yield high regret. We then propose a simple and computationally efficient hybrid search strategy which outperforms the existing approaches. We highlight the practical benefits of the proposed solution on a set of 19 diverse vision tasks.

## 1 INTRODUCTION

Services such as TensorFlow Hub[1] or PyTorch Hub[1] offer a plethora of pre-trained models that often achieve state-of-the-art performance on specific tasks in the vision and NLP domains. The predominant approach, namely choosing a pre-trained model and *fine-tuning* it to the downstream task, remains a very strong and data efficient baseline. This approach is not only successful when the pre-training task is similar to the target task, but also across tasks with seemingly differing characteristics, such as applying an ImageNet pre-trained model to medical applications like diabetic retinopathy classification (Oquab et al., 2014). Fine-tuning often entails adding several more layers to the pre-trained deep network and tuning all the parameters using a limited amount of downstream data. Due to the fact that all parameters are being updated, this process can be extremely costly and intensive in terms of compute (Zhai et al., 2019). Fine-tuning all models to find the best performing one is rapidly becoming computationally infeasible. A more efficient alternative is to simply train a linear classifier or a $k$-nearest neighbour ($kNN$) classifier on top of the learned representation (e.g. pre-logits). However, the performance gap with respect to fine-tuning can be rather large (Kolesnikov et al., 2019; Kornblith et al., 2019).

In this paper we study the application of *computationally efficient methods for determining which model(s) one should fine-tune for a given task at hand*. We divide existing methods into two groups: (a) *task-agnostic* model search strategies – which rank pre-trained models independently of the downstream task (e.g. sort by ImageNet accuracy, if available), and (b) *task-aware* model search strategies – which make use of the provided downstream dataset in order to rank models (e.g. $kNN$ classifier accuracy as a proxy for fine-tuning accuracy) (Kornblith et al., 2019; Meiseles & Rokach, 2020; Puigcerver et al., 2020).

Clearly, the performance of these strategies depends on the set of models considered and the computational constraints of the practitioner (e.g. the memory footprint, desired inference time, etc.). To this end, we define several *model pools* and study the performance and generalization of each strategy across different pools. In particular, we make sure that these pools contain both "generalist" models (e.g. models trained on ImageNet), but also "expert" models (e.g. models trained on domain-specific datasets, such as flowers, animals, etc.).

---

[1]`https://tfhub.dev` and `https://pytorch.org/hub`

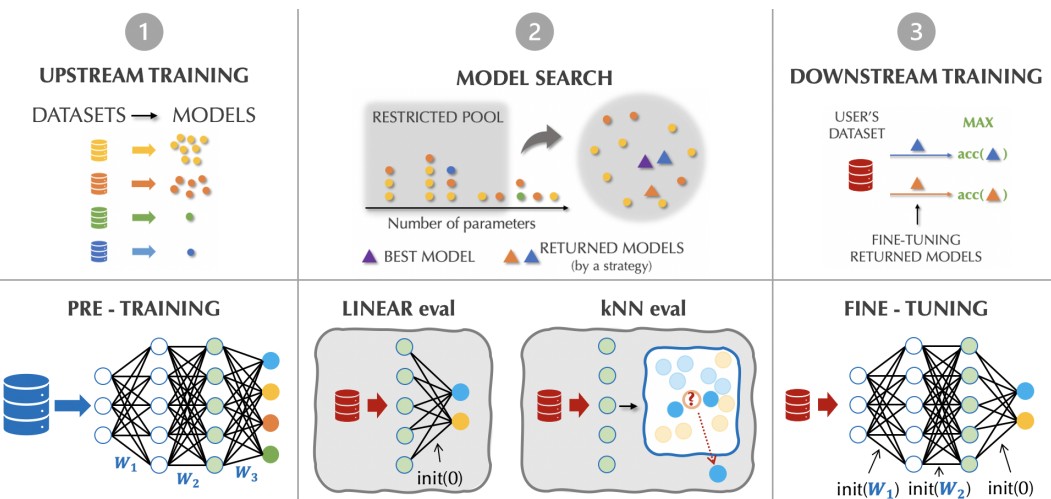

Figure 1: Transfer learning setup: **(1) Upstream models** Pre-training of models from randomly initialized weights on the (large) upstream datasets; **(2) Model search** Either downstream task independent or by running a proxy task, i.e. fixing the weights of all but the last layer and training a *linear classifier* or deploying a *kNN classifier* on the downstream dataset; **(3) Downstream training** Unfreezing all the weights, optimizing both the pre-defined ones and a new linear classification layer on the downstream dataset.

**Our contributions.** (i) We formally define and motivate the model search problem through a notion of regret. We conduct the first study of this problem in a realistic setting focusing on heterogeneous model pools. (ii) We perform a large-scale experimental study by fine-tuning 19 downstream tasks on 46 models from a heterogeneous set of pre-trained models split into 5 meaningful and representative pools. (iii) We highlight the dependence of the performance of each strategy on the constrained model pool, and show that, perhaps surprisingly, both task-aware and task-agnostic proxies fail (i.e. suffer a large regret) on a significant fraction of downstream tasks. (iv) Finally, we develop a hybrid approach which generalizes across model pools as a practical alternative.

## 2    BACKGROUND AND RELATED WORK

We will now introduce the main concepts behind the considered transfer learning approach where the pre-trained model is adapted to the target task by learning a mapping from the intermediate representation to the target labels (Pan & Yang, 2009; Tan et al., 2018; Wang, 2018; Weiss et al., 2016), as illustrated in Figure 1.

**(I) Upstream models.**    Upstream training, or *pre-training*, refers simply to a procedure which trains a model on a given task. Given the variety of data sources, losses, neural architectures, and other design decisions, the set of upstream models provides a diverse set of learned representations which can be used for a downstream task. In general, the user is provided with these models, but can not control any of these dimensions, nor access the upstream training data. Previous work call the models in these pools *specialist* models (Ngiam et al., 2018) or *experts* (Puigcerver et al., 2020).

**(II) Model search.**    Given no limits on computation, the problem is trivial – exhaustively fine-tune each model and pick the best performing one. In practice, however, one is often faced with stringent requirements on computation, and more efficient strategies are required. The aim of the second stage in Figure 1 is therefore to select a small number of models from the pool, so that they can be fine-tuned in the last step. The central research question of this paper is precisely how we can choose the most promising models for the task at hand. We divide existing related work and the methods we focus on later in this work into three categories as illustrated in Figure 2.

**(II A) Task-agnostic search strategies.**    These strategies rank models without looking at the data of the downstream task (Kornblith et al., 2019). As a consequence, given a fixed pool of candidates, the same model is recommended for *every* task. We focus on the following popular task-agnostic approach: (i) Pick the highest test accuracy ImageNet model (if such a model is in the pool), otherwise (ii) pick the one trained on the largest dataset. If there is a tie, pick the biggest model in the pool (in terms of the number of parameters).

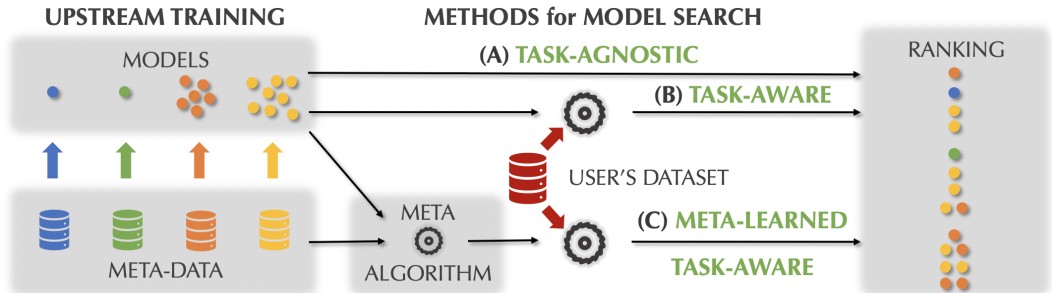

Figure 2: Model search methods: **(A) Task-agnostic** methods do not see the downstream task, producing the same ranking of models for all possible tasks (e.g., using the highest ImageNet accuracy); **(B) Task-aware** methods deploy a proxy (e.g., linear evaluation) for each model on user's dataset, without referring to meta data used during pre-training; **(C) Meta-learned task-aware** methods deploy a proxy for each model on user's dataset, together with meta data used during pre-training. For example, one could use some notion of similarity between tasks, as in Achille et al. (2019).

**(II B) Task-aware search strategies.** In contrast to the task-agnostic approach, task-aware methods use the downstream data, thus requiring additional computation. In this work we first compute and freeze the representations extracted from the last layer, noting that this step requires a forward pass (i.e., running inference) for each example on every model in the pool. On top of that, one usually uses a classification method to solve the task with the new features provided by the pre-trained model. We choose two classifiers that are commonly used for this purpose: a linear and a $k$NN classifier. We note that both are usually several orders of magnitude faster than fine-tuning the whole model, with $k$NN being a non-parametric classifier that can capture non-linear relationships. Finally, we select the models that achieve highest accuracy in this simplified setup.

Alternative approaches such as LEEP (Nguyen et al., 2020), NCA (Tran et al., 2019) or H-Score (Bao et al., 2019) replace the classifier's accuracy by a somewhat cheaper estimator. These three approaches have a clear limitation in supporting only those cases in which a classification head for the original task is provided together with each pre-trained model. Furthermore, the theoretical guarantees are given with respect to models trained on top of the frozen representations only. As observed with LEEP, albeit on natural datasets only, fine-tuning achieves significantly better test accuracies than a linear classifier, which further invites for better understanding of using linear accuracy as a proxy for fine-tuning, noting that linear correlations are not necessarily transitive. Furthermore, training a single linear layer, or calculating the kNN accuracy on top of transformed representations, is typically much less computational demanding than running the inference itself, and both steps can easily be overlapped. Recently, Meiseles & Rokach (2020) introduced the Mean Silhouette Coefficient (MSC) to forecast a model's performance after fine-tuning on time series data. We omit this approach due to its provable relation to a linear classifier proxy, whilst being similar to $k$NN with respect to capturing non-linear relationships. Finally, Puigcerver et al. (2020) utilize $k$NN as a cheap proxy task for searching models in a set of experts with the same architecture.

**(II C) Meta-learned task-aware search strategies.** In the last category, we allow model search strategies to utilize meta-data about the pre-trained models in order to more efficiently rank models. Such strategies typically contain a meta-learning part that precedes the actual search component. The meta-learned dependencies between models can be used in a computational (Achille et al., 2019) or a semantical (Zamir et al., 2018; Song et al., 2020; Dwivedi et al., 2020) way. Model2Vec, a generalization of Task2Vec Achille et al. (2019), learns an embedding for each pre-trained model as a combination of a task vector and model bias. It minimizes a cross-entropy loss based on the accuracy achieved on the source dataset by the corresponding models. Given a new task, the model search part is performed by finding the nearest model embedding given the computed task embedding for the downstream task. This step is computationally cheap as it requires training only a single probe-network in order to compute the new task embedding. Therefore, the overall compute time resides with the owner of model repositories. We omit Model2Vec for the remainder of the paper due to limited understanding and computational burden of the incremental re-training when adding a new pre-trained model to the pool, and the less practical use case for the current users of available pre-trained model repositories. Having said that, in the final discussion section of this work we address how Model2Vec could benefit from our proposed strategies, hoping that this could open an interesting and fruitful followup research area.

**(III) Downstream training.** In this stage, the selected model is adapted to the downstream task (cf. Figure 1). The predominant approach is to fully or partially apply the pre-trained neural network as a feature extractor. The head (e.g. last linear layer) of the pre-trained model is replaced with a new one, and the whole model is trained on the target data. This process is commonly referred to as *fine-tuning* and it often outperforms other methods (Oquab et al., 2014; Donahue et al., 2014; Sharif Razavian et al., 2014; Kornblith et al., 2019).

## 3  COMPUTATIONAL BUDGET AND REGRET

The main aim of this work is the study of simple methods to filter and search pre-trained models before stepping into the –more expensive– fine-tuning process. Formally, we define a search method $m(\mathcal{M}, \mathcal{D})$ with budget $B$ as a function which takes a set of models $\mathcal{M}$ and a downstream dataset $\mathcal{D}$ as input, and outputs a number of distinct models $\mathcal{S}_m \subseteq \mathcal{M}$, with $|\mathcal{S}_m| = B$. Those $B$ models are then all fine-tuned in order to obtain the best possible test accuracy on the downstream task $\mathcal{D}$.

**Budget and regret.** Fine-tuning represents the largest computational cost; accordingly, we define the *number* of models that are fine-tuned as the computational complexity of a given method. Given any fixed budget $B$, we would like to return a set $\mathcal{S}$ which includes the models resulting in good performance downstream. In particular, we define the notion of *absolute regret* of a search strategy $m$ and a pool of models $\mathcal{M}$ on dataset $\mathcal{D}$ as

$$\underbrace{\max_{m_i \in \mathcal{M}} \mathbf{E}[t(m_i, \mathcal{D})]}_{\text{ORACLE}} - \underbrace{\mathbf{E}\left[\max_{s_i \in \mathcal{S}_m} t(s_i, \mathcal{D})\right]}_{s(m)}, \tag{1}$$

where $t(m, \mathcal{D})$ is the test accuracy achieved when fine-tuning model $m$ on dataset $\mathcal{D}$. The first expectation is taken over the randomness in the $t(\cdot)$ operator, that is, the randomness in fine-tuning and due to a finite sampled test set. In addition to the randomness in $t(\cdot)$, the second expectation also accounts for any potential randomization in the algorithm that computes $\mathcal{S}_m$. We define $s(m)$ as the expected maximal test accuracy achieved by any model in the set $\mathcal{S}_m$, the set of models returned by a fixed strategy $m$. In our case, $k$NN is deterministic as all the downstream data is used, whereas the linear model depends on the randomness of stochastic gradient descent. To enable comparability between datasets of different difficulty as well as a comparison between two selection strategies $m_1$, and $m_2$, we define their *relative delta* as

$$\Delta(m_1, m_2) := \frac{s(m_1) - s(m_2)}{1 - \min(s(m_1), s(m_2))}, \tag{2}$$

with $s(\cdot) \in [0, 1]$ as defined in Equation 1. Substituting $s(m_1)$ by the ORACLE value, and $s(m_2)$ by $s(m)$ leads to the *relative regret* $r(m)$. We discuss the impact of alternative notions in Section 5.4.

## 4  EXPERIMENTAL DESIGN

Our goal is to assess which model search strategies achieve low regret when presented with a diverse set of models. As discussed, there are three key variables: (i) The set of downstream tasks, which serve as a proxy for computing the expected regret of any given strategy, (ii) the *model pool*, namely the set we explore to find low-regret models, and (iii) the transfer-learning algorithms.

### 4.1  DATASETS AND MODELS

**Datasets.** We use VTAB-1K, a few-shot learning benchmark composed of 19 tasks partitioned into 3 groups – ●*natural*, ●*specialized*, and ●*structured* (Zhai et al., 2019). The *natural* image tasks include images of the natural world captured through standard cameras, representing generic objects, fine-grained classes, or abstract concepts. *Specialized* tasks contain images captured using specialist equipment, such as medical images or remote sensing. The *structured* tasks are often derive from artificial environments that target understanding of specific changes between images, such as predicting the distance to an object in a 3D scene (e.g. DeepMind Lab), counting objects (e.g. CLEVR), or detecting orientation (e.g. dSprites for disentangled representations). Each task has 800 training examples, 200 validation examples, and the full test set. This allows us to evaluate model search strategies on a variety of tasks and in a setting where transfer learning offers clear benefits with respect to training from scratch (Zhai et al., 2019).

**Models.** The motivation behind the model pools is to simulate several use-cases that are ubiquitous in practice. We first collect 46 classification models (detailed overview in Appendix A):

- 15 models trained on the ILSVRC 2012 (ImageNet) classification task (Russakovsky et al., 2015) including Inception V1-V3 models (Szegedy et al., 2016), ResNet V1 and V2 (depth 50, 101, and 152) (He et al., 2016), MobileNet V1 and V2 (Howard et al., 2017), NasNet (Zoph et al., 2018) and PNasNet (Liu et al., 2018) networks.

- 16 ResNet-50-V2 models trained on (subsets of) JFT (Puigcerver et al., 2020). These models are trained on different subsets of a larger dataset and perform significantly better on a small subset of downstream tasks we consider (i.e. they can be considered as *experts*).

- 15 models from the VTAB benchmark[2], with a diverse coverage of losses (e.g. generative, self-supervised, self-supervised combined with supervised, etc.) and architectures. In all cases the upstream dataset was ImageNet, but the evaluation was performed across the VTAB benchmark which does not include ImageNet.

## 4.2 Model pools

**(A) Identifying good resource-constrained models (RESNET-50, DIM2048).** Here we consider two cases: (i) RESNET-50: All models with the number of parameters smaller or equal to ResNet50-V2. While the number of parameters is clearly not the ideal predictor, this set roughly captures the models with limited memory footprint and inference time typically used in practice. Most notably, this pool excludes the NasNet and PNasNet architectures, and includes the *expert* models. (ii) DIM2048: The transfer strategies discussed in Section 2 might be sensitive to the size of the representation. In addition, restricting the representation size is a common constraint in practical settings. Hence, we consider a model pool where the representation dimension is limited to a maximum of 2048.

**(B) Identifying expert models in presence of non-experts (EXPERT).** We consider a pool of 16 ResNet-50-V2 models from Puigcerver et al. (2020). These models which we considered as *experts* are trained on different subsets of a larger dataset. As the number of available models and the upstream training regimes increase, the number of such experts is likely to increase. As such, this presents a realistic scenario in which an expert for the target task may be present, but it is hard to identify due to the presence of other models, some of which might perform well *on average*.

**(C) Do better task-agnostic models transfer better (IMNETACCURACIES)?** This pool offers the ability to choose an upstream representation-learning technique that is best suited for a specific downstream task. This pool is mainly used to validate the idea that (a) ImageNet models transfer well across different tasks (Huh et al., 2016; He et al., 2019) and that (b) better ImageNet models transfer better (Kornblith et al., 2019).

**(D) All models (ALL).** Finally, we consider the hardest setting, namely when the model pool contains all 46 models and no conceptual nor computational restrictions are in place. We note that: EXPERT $\subset$ RESNET-50 $\subset$ DIM2048 $\subset$ ALL and IMNETACCURACIES $\subset$ ALL.

## 4.3 Evaluation procedures

**Fine tuning.** To assign a downstream test accuracy to each pair of model and task, we use the median test performance of 5 models obtained as follows: (i) Add a linear layer followed by a softmax layer and train a model on all examples of the training set. (ii) Fine-tune the obtained model twice, considering two learning rates, and 2500 steps of SGD and a batch size of 512 (Zhai et al., 2019). (iii) Return the model with the highest validation accuracy. Note that in this case, the entire model, and not only the linear layer, is retrained. As a result, there are 10 runs for each model and we obtain 8740 trained models ($46 \times 19 \times 5 \times 2$).

**Linear evaluation.** We train a logistic regression classifier added to the model representations (fixed) using SGD. We consider two learning rates (0.1 and 0.01) for 2500 steps and select the model with the best validation accuracy. For robustness we run this procedure 5 times and take the median validation accuracy out of those resulting values. As a result, we obtain again 8740 models.

*k***NN evaluation.** We compute the validation accuracy by assigning to each of the 200 validation samples the label of the nearest training example (i.e. $k = 1$) using standard Euclidean distance.

---

[2]https://tfhub.dev/vtab

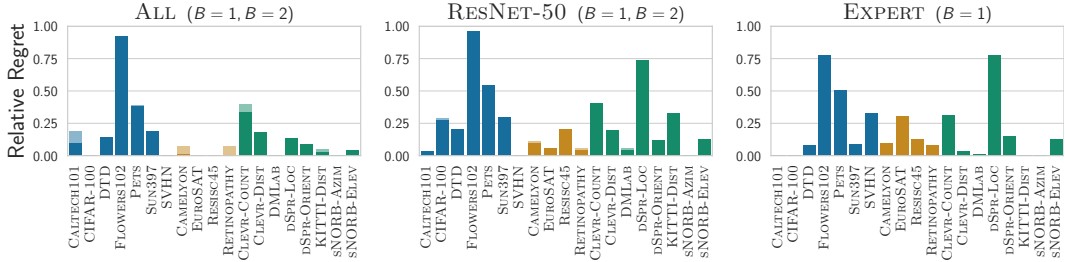

Figure 3: **Task-agnostic strategies.** Relative regret ($r(m)$, cf. Section 3) with $B = 1$ (transparent) and $B = 2$ (solid) on the ALL, RESNET-50 and EXPERT pools, bearing in mind that there is only one task-agnostic model in EXPERT. By definition, task-agnostic strategies exclude experts yielding high regret on the RESNET-50 and EXPERT pools, particularly on natural or structured datasets.

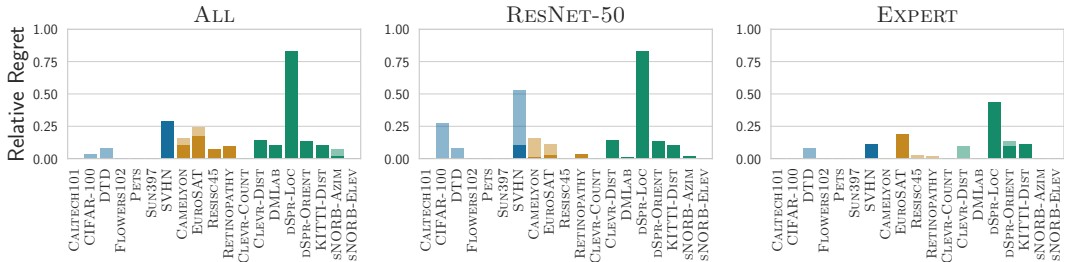

Figure 4: **Task-aware strategies (linear).** Relative regret for $B = 1$ (transparent) and $B = 2$ (solid) on the ALL, RESNET-50, and EXPERT pools. Compared to task-agnostic strategies, we observe improvement on natural datasets (except SVHN) and on restricted pools (except DSPR-LOC), due to its ability to properly choose experts.

## 5 KEY EXPERIMENTAL RESULTS

In this section we challenge common assumptions and highlight the most important findings of this study, whilst the extended analysis containing all the plots and tables can be found in the supplementary material. We remark that in the main body we only consider three main pools of models – ALL, RESNET-50 and EXPERT, as we see them as the most representative ones. Since DIM2048 behaves very similarly to RESNET-50, whereas IMNETACCURACIES is used only to confirm the findings of Kornblith et al. (2019), the results of ablation studies involving these two pools can be found in Appendix D. Finally, in this section we mainly investigate linear evaluation as the task-aware choice; all the corresponding plots for $k$NN can be found in Appendix E.

### 5.1 HIGH REGRET OF TASK-AGNOSTIC STRATEGIES

Figure 3 shows the results for task-agnostic methods with budget $B = 1$ and $B = 2$ on the ALL, RESNET-50, and EXPERT pools. We observe a significant regret, in particularly for RESNET-50 and EXPERT pools (30% of the datasets have a relative regret larger than 25% on those two pools). This highlights the fact that task-agnostic methods are not able to pick expert models, in particular on natural and structured datasets. As more experts become available, this gap is likely to grow, making it clear that task-agnostic strategies are inadequate on its own.

### 5.2 ARE TASK-AWARE ALWAYS A GOOD PREDICTOR?

Intuitively, having access to the downstream dataset should be beneficial. We evaluate both the linear and the $k$NN predictor as detailed in Section 4. Figure 4 provides our overall results for the linear model, whereas analogous results for $k$NN are presented in Appendix E. The method struggles on some structured datasets (in particular on DSPR-LOC).

Compared to the task-agnostic strategy, as presented in Figure 5 for $B = 1$, we observe significant improvements on restricted model pools. The EXPERT pool benefits the most: linear evaluation

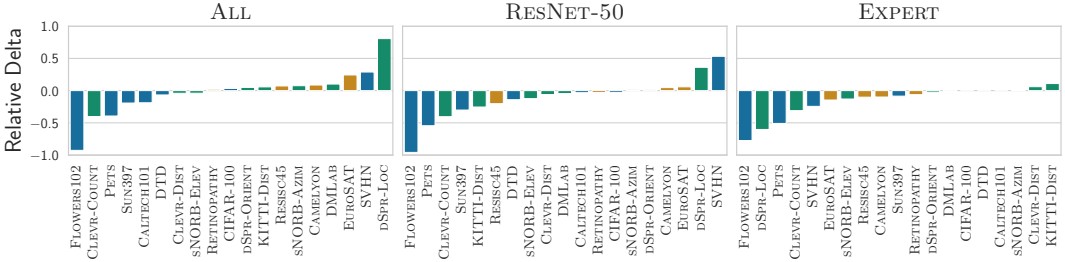

Figure 5: **Task-agnostic** (positive if better) **vs Task-aware (linear)** (negative if better) for $B = 1$. On the ALL pool, the methods perform in a similar fashion, with respect to the number of datasets and the amount in which one outperforms the other. When one restricts the pool to RESNET-50 or EXPERT task-aware methods outperform the task-agnostic method on most datasets. The relative delta is defined in Equation 2 in Section 3.

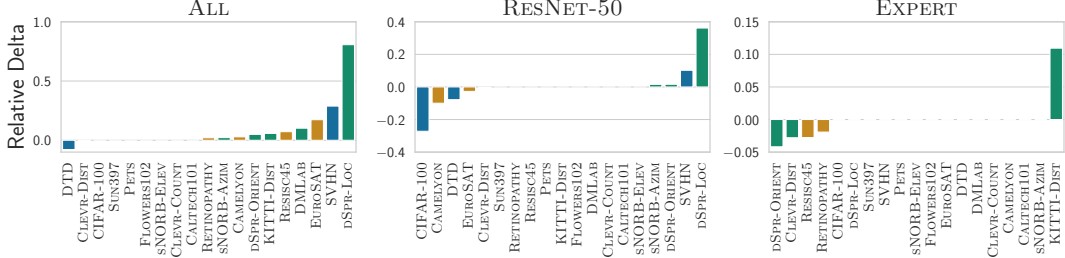

Figure 6: **Hybrid linear** (positive if better) **vs Linear evaluation** (negative if better) for $B = 2$. We observe that hybrid linear significantly outperforms linear with the same budget on the ALL pool. Even though for RESNET-50 and EXPERT pools there are datasets on which linear performs better than hybrid, the amounts in which it does are usually small. We note that most significant gains of hybrid come on certain structured datasets, the hardest task for every strategy.

outperforms task-agnostic methods on almost every dataset (task-aware is only outperformed on three datasets by more than 1%, and by 10% in the worst case on the KITTI-DIST dataset). On the other hand, task-agnostic and task-aware strategies seem to outperform each other on a similar number of datasets and by a comparable magnitude in the ALL pool. This suggests that no single strategy uniformly dominates all other strategies across pools.

In order to understand this further, we perform an ablation study where we plot the linear and $k$NN regret on the IMNETACCURACIES pool in Appendix D. In Figures 15 and 16 we observe that task-aware search methods perform rather poorly when having access *only* to different architectures trained on the same upstream data. The IMNETACCURACIES models are included in the ALL pool, and in some datasets some of those models are the best-performing ones.

Performance of the $k$NN predictor is on par on half of the datasets across the pools, and slightly worse than linear evaluation on the other half. We present these findings in Figure 20 in Appendix E.

### 5.3 HYBRID APPROACH YIELDS THE BEST OF BOTH WORLDS

A hybrid approach that selects both the top-1 task-agnostic model and the top-$(B - 1)$ task-aware models leads to strong overall results. Figure 6 shows how the hybrid approach with linear evaluation as the task-aware method significantly outperforms its linear counterpart with $B = 2$. This is most noticeable in the ALL pool where the task-agnostic model provides a large boost on some datasets.

As we saw in Figure 5, when looking at the ALL pool, the task-agnostic candidate tends to beat the linear one on datasets such as DSPR-LOC, SVHN or EUROSAT. Similarly, the linear candidate model clearly outperforms its task-agnostic counterpart on many natural datasets such as FLOWERS or PETS. A comparison of Figures 5 and 6 reflects how the dominance of linear-only strategy vanishes on most datasets when confronted with the hybrid approach. For the RESNET-50 and EXPERT pools, as expected, the hybrid algorithm preserves the good picks of the linear proxy. That said, we observe an increase of 36% on DSPR-LOC in the RESNET-50, and 11% on KITTI-DIST. Both are structured datasets on which the linear proxy task performs poorly, as shown in Figure 4.

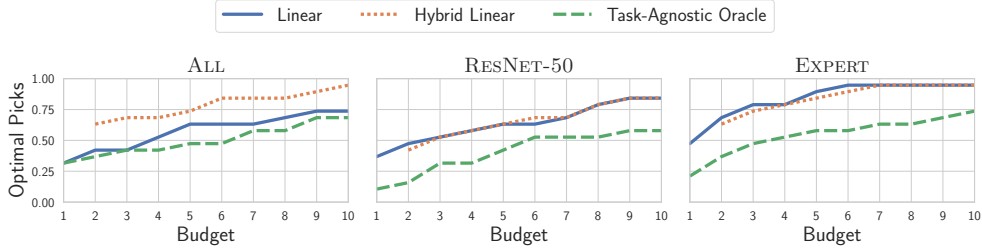

Figure 7: **Optimal picks as a function of the computational budget.** The number of picked models (relative) with zero regret across three representative pools. We note that hybrid linear outperforms all other methods on ALL, whilst being comparable with the linear strategy on restricted pools where linear alone already performs well. Here, the task-agnostic oracle refers to a method which ranks models based on their average accuracy across all datasets (more details Section 5.4).

The hybrid strategy requires to fine-tune at least two models. Given that it performs well across all model pools and datasets, this is a reasonable price to pay in practice, and we suggest its use as the off-the-shelf approach. Figures 21 and 22 in Appendix E depict the results for $k$NN. In EXPERT models, the second $k$NN pick tends to beat the task-agnostic one – hurting the $k$NN hybrid outcomes. Overall, the hybrid linear approach consistently outperforms the one based on $k$NN.

### 5.4 FURTHER ABLATION STUDIES

**How does the computational budget impact the findings?** We have seen that for a limited budget of $B = 2$ the proposed hybrid method outperforms the other strategies. A natural question that follows is: how do these methods perform as a function of the computational budget $B$? In particular, for each budget $B$, we compute how frequently does a strategy pick the best model. The results are shown in Figure 7. We observe that the hybrid linear strategy outperforms all individual strategies on the ALL pool. Furthermore, it also outperforms a strong impractical task-agnostic oracle which is allowed to rank the models by the average fine-tune accuracy over all datasets. Our hybrid strategy achieves an on par performance with the linear approach on pools on which linear performs well. When task-aware strategies perform badly (e.g. pools without expert models), hybrid linear is significantly stronger (cf. Figure 17 in Appendix D). These empirical results demonstrate that the hybrid strategy is a simple yet effective practical choice.

**Alternative evaluation scores.** Both Meiseles & Rokach (2020) and Kornblith et al. (2019) compute the correlation between the ImageNet test accuracy and the average fine-tune accuracy across datasets. Although this provides a good task-agnostic evaluation method for the *average performance* (cf. Figure 1, right, in Kornblith et al. (2019)), it can be significantly impacted by outliers that have poor correlations on a specific dataset (cf. Figure 2, middle row, in Kornblith et al. (2019)). In Appendix B we highlight another limitation of using correlation scores in the context of model-search strategies across heterogeneous pools. Nevertheless, we empirically validate that ranking the models based on their ImageNet test accuracy on the IMNETACCURACIES pool transfers well to our evaluation setting (cf. Figure 14 in the Appendix D). Furthermore, we show that reporting the differences of logit-transformed accuracies (log-odds) leads to similar conclusions as ours (cf. Appendix C). We opt for the relative regret $r(m)$, defined in Section 3, as it is more intuitive and contained in $[-1, 1]$.

**Impact of the $k$NN hyperparameters.** The $k$NN classifier suffers from the curse of dimensionality (Snapp et al., 1991), which is why we study the impact of the dimension (i.e. the representation size) on the $k$NN evaluation. We fix a dataset, and plot a model's $k$NN score versus its representation dimension. In order to have a single point per dimension and avoid an over-representation of the expert models that are all of the same architecture, we choose the model with the best $k$NN accuracy. By calculating the Pearson correlation coefficient between the dimension and the respective $k$NN scores, we observe a moderate anti-correlation ($R < -0.5$) for 3 datasets, a moderate correlation ($R > 0.5$) for 3 other datasets, and either small or no correlation for the remaining 13 datasets. Based on this empirical evidence we conclude that there is no significant correlation between the $k$NN classifier accuracy and the dimension. We provide more details in Figure 23 of Appendix F. Regarding $k$, our preliminary experiments with $k = 3$ offered no significant advantages over $k = 1$.

## 6 OTHER RELATED WORK

Given access to the meta-data such as the source datasets, one could compare the upstream and downstream datasets (Bhattacharjee et al., 2020); blend the source and target data by reweighting the upstream data to reflect its similarity to the downstream task (Ngiam et al., 2018); or construct a joint dataset by identifying subsets of the upstream data that are well-aligned with the downstream data (Ge & Yu, 2017). These approaches are restricted to one model per upstream dataset, while also being less practical as they necessitate training a new model as well as access to upstream datasets which might be unavailable due to proprietary or privacy concerns.

Another branch of related work is given by Zamir et al. (2018). Taskonomy provides an approach that finds the best source dataset(s) by exhaustively exploring the space of all possibilities. However, it is not-comparable to our setting for two reasons: (i) the input domain and data is assumed to remain constant and task are only different in their labels, (ii) the chosen architecture keeps the weights of the encoder frozen, and the decoder trained on top usually consists of multiple fully connected layers, opposed to our fine-tuning regime with a single linear layer on top. Improvements by Song et al. (2020) and Dwivedi et al. (2020) make taskonomy faster, but they still do not distinguish multiple models trained on the same dataset, nor bypass the constraints described previously.

Finally, best-arm identification bandits algorithms suggest the successive elimination of sub-optimal choices (Even-Dar et al., 2006) derived by fine-tuning models for shorter time. This combines model selection and downstream training, noting that its computation and theoretical properties heavily depend on the architectures of pre-trained models as well as hyper-parameters used therein (e.g., batch norms), which is still far from being understood by the community. Therefore, this line of work including partial fine-tuning as a proxy task is deliberately omitted in this work.

## 7 CONCLUSIONS, LIMITATIONS, AND FUTURE WORK

Transfer learning offers a data-efficient solution to train models for a range of downstream tasks. As we witness an increasing number of models becoming available in repositories such as TensorFlow Hub, though, finding the right pre-trained models for target tasks is getting harder. Fine-tuning all of them is not an option. In practice, the computational budget is limited and efficient model search strategies become paramount. We motivate and formalize the problem of efficient model search through a notion of *regret*, and argue that regret is better suited to evaluate search algorithms than correlation-based metrics. Empirical evaluation results for the predominant strategies, namely *task-agnostic* and *task-aware* search strategies, are presented across several scenarios, showing that both can sometimes yield high regret. For any individual method we study, there exists a pool of models on which the method fails. Finally, we propose a simple and computationally efficient hybrid search strategy which consistently outperforms the existing approaches over 19 diverse vision tasks and across all the defined model pools.

**Limitations and future work.** To further stress-test the generalization of analysed strategies, the number of relevant model pools could be increased by incorporating more diverse upstream tasks, in particular neural architectures, losses, and datasets. This would potentially yield more expert models making the task even more challenging, and it could further highlight the advantages of an effective search strategy. Secondly, we observe that task-aware methods consistently perform poorly in specific cases, such as when we consider diverse architectures trained only on ImageNet. There is no obvious reason for such failures. Similarly, there seems to be a clear pattern where task-aware methods perform significantly worse on structured datasets than on natural ones. We hypothesise that this is due to the lack of adequate expert models for these domains. However, an in-depth analysis of these specific cases might be beneficial and insightful. Thirdly, despite the fact that we did not include meta-learned task-aware search strategies on purpose, we believe that the hybrid strategy would significantly improve the performance of Model2Vec, and hence Task2Vec (Achille et al., 2019), or other meta-learned task-aware search strategies in general. For example, in Figure 3 in Achille et al. (2019) the expert selection procedure is outperformed by the simple generalist approach in 15 out of 50 tasks, sometimes by a large margin.

Finally, transfer learning is a successful strategy in various natural language processing tasks, which we did not explore here due to the lack of diverse sources of text representation and heterogeneous pools in online repositories. Going beyond vision tasks presents a very clear research direction.

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
