# OpenReview forum: "Which Model to Transfer? Finding the Needle in the Growing Haystack"
_ICLR.cc/2021/Conference — Reject_

### Official Review · AnonReviewer2 · 2020-10-17
**A Hybrid (Ensemble) Approach for Pretrained Model Search**

**Rating:** 4
**Confidence:** 3

**Review:**

### Summary

The paper evaluates three procedures for selecting models for transfer learning. The choices are task-agnostic selection, linear training, and the hybrid approach. They empirically show that the hybrid algorithm works the best on few-shot learning on images.

### Feedback

* The paper is a straightforward paper and easily understandable. The message is practical, but not very surprising. Unfortunately, it is only on image data; it would have been great if the authors had used an example from NLP too.
* The hybrid approach is super-simple, which is nice. The results in Figure 6 confirm that how its ensemble nature helps. Although it does not necessarily outperform the linear algorithm in Figure 6. The ensembling approach also does not seems to be the optimal solution. The authors could study the generalization performance of the hybrid algorithm to provide further insights.
* An empirical run-time analysis is missing.
* While the authors indicate that all models perform comparatively poorly on the structured tasks, they do not provide specific insights about the root cause of this.
* Overall, the idea is simple and practical, but the methodological contributions of this paper is rather limited.

--------
### Post-Response Update
Unfortunately, the authors' response is not satisfactory on multiple issues. Thus, I reduce my rating by one point.

---

> ### Author Response · Authors · 2020-11-19
> **Response to AnonReviewer2**
>
> We thank the reviewer for the thoughtful comments and are grateful to see that the simple yet consistent proposed solution is appreciated.
>
> _[Unfortunately, it is only on image data; it would have been great if the authors had used an example from NLP too.]_
>
> We agree that extending this work to include NLP tasks is an important followup project. At this stage, fine-tune strategies on its own are less explored and unclear in NLP compared to vision tasks (e.g., pre-trained models are typically extended by more than one linear classification layer, or the representations are not always taken from the last pre-logit layer).
>
> _[Although it does not necessarily outperform the linear algorithm in Figure 6.]_
>
> We see Figure 6 (which is now Figure 7 in the revised version) as a confirmation of some of our claims, rather than a weakness:
>
>
>
> *   Since hybrid with budget B is based on linear on budget B-1, it is **expected that the graphs are fairly similar when experts are the best models** (which is the case in restricted pools)
> *   It shows the **necessity of including the task-agnostic method** - on ALL pool we clearly see that linear is not able to choose the best model
> *   We want to emphasize that Figure 6 cares only about winners, which differs from the rest of the paper where we look at the notion of regret
>
> _[The ensembling approach also does not seems to be the optimal solution. The authors could study the generalization performance of the hybrid algorithm to provide further insights.]_
>
> The generalization property of the hybrid strategy using a budget of B, by definition, comes directly from the generalization of the task-aware strategy for B-1 and the first pick of the task-agnostic strategy. Giving any generalization bounds for training or fine-tuning deep neural networks is already a hard and currently open research problem, which we do not aim at solving here. Instead, we showcase empirical failures of each strategy which, interestingly, do not overlap much, enabling consistent improvements using the proposed hybrid strategy.
>
> _[An empirical run-time analysis is missing.]_
>
> Efficient implementation of all methods, which is crucial for a fair comparison, is out of the scope of this work. At this stage, understanding the limitations of using task-aware (especially a linear classifier) or task-agnostic search methods is the founding block. Run-time analysis, extension towards related work that provides cheaper estimators (such as LEEP (Nguyen et al., 2020),  NCA (Tran  et  al.,  2019) or H-Score (Bao et al., 2019)), in order to speed up the entire search process, are all valid future projects that our work opens.

---

### Official Review · AnonReviewer4 · 2020-10-19
**Good baselines for model selection, but paper ignores many prior papers on this problem**

**Rating:** 6
**Confidence:** 5

**Review:**

Paper summary: This paper looks at the problem of efficiently choosing pre-trained models as initialization for downstream target tasks. It compares 3 strategies, a task-agnostic one which uses imagenet accuracies, a task-aware one which uses the acccuracy of linear classifiers on fixed representations, and a hybrid one which combines the two.

Pros:
+ The evaluation is fairly thorough. I especially like the fact that the authors consider the different axes along which pre-trained models differ (model capacity, generalist/experts etc.)
+ The pool of downstream datasets is large.
+ The suggested strategy is simple and easy to implement.
+ The problem is significant in practice since almost all practical applications of neural networks have this prroblem, and the gains seem large. I wish there was more work on this problem.

Cons:
- The biggest issue is that this paper ignores several important papers publiished before on this problem. Especially of note is the Task2Vec approach, which computes model and task embeddings. I would like comparisons both in terms of accuracy/regret as well as computational cost:
Alessandro Achille, Michael Lam, Rahul Tewari, Avinash Ravichandran, Subhransu Maji, Charless C. Fowlkes, Stefano Soatto, Pietro Perona; Proceedings of the IEEE/CVF International Conference on Computer Vision (ICCV), 2019, pp. 6430-6439

Other papers that are also relevant and should be cited and comparisons discussed:
Bishwaranjan Bhattacharjee, John R. Kender, Matthew Hill, Parijat Dube, Siyu Huo, Michael R. Glass, Brian Belgodere, Sharath Pankanti, Noel Codella, Patrick Watson; Proceedings of the IEEE/CVF Conference on Computer Vision and Pattern Recognition (CVPR) Workshops, 2020, pp. 760-761

Amir R. Zamir, Alexander Sax, William Shen, Leonidas J. Guibas, Jitendra Malik, Silvio Savarese; Proceedings of the IEEE Conference on Computer Vision and Pattern Recognition (CVPR), 2018, pp. 3712-3722

- The approach is not particularly novel. There is also no novel technical insight that explains the results.

- The use of the JFT dataset hampers reproducibility since the dataset is not public. I'd like to see results with JFT excluded.

For acceptance, I would definitely want to see the first of these convincingly addressed.
[Updated rating]

---

> ### Author Response · Authors · 2020-11-19
> **Response to AnonReviewer4**
>
> We thank the reviewer for the thoughtful comments and pointing out the potentially relevant related work. We have updated the main body of the paper, in particular the "Background" section and the "Other related work" section to improve the positioning of our work, and added Figure 2 for further clarification. That being said, we believe that our contributions are relevant to the research community and the practitioners.
>
> [Related methods]
>
>
>
> *   Task2Vec (and Model2Vec) (Achille et. al.):
>     *   We highlight multiple aspects when comparing “meta-learned task-aware” strategies such as Model2Vec to our choice for search strategies (also described in the revised paper)
>     *   [Computational comparison] Assuming access to M models, our proxy tasks certainly require more compute time than Model2Vec, O(M) compared to O(1) for the search part. However, **the computational requirement for Model2Vec is shifted to the meta-learn algorithm** (which needs to be rerun from scratch whenever new pre-trained models are available - a **different category of model search strategies than what we examine**), which has an asymptotic complexity of O(M) for Model2Vec.
>     *   [Hybrid helps Model2Vec] Figure 3 in the Task2Vec paper shows that the **generalist model outperforms the selected expert in 15 out of 50 cases**, sometimes by a large margin - **hybrid strategy should improve their proposed method significantly**, formulating a very interesting future research problem.
> *   P2L (Bhattacharjee et. al.):
>     *   Estimates the impact of transferring the learned representations from a source dataset (unclear how one could distinguish multiple models trained on the same dataset) by incorporating the dataset size and the divergence between the upstream and downstream dataset.
>     *   This is **orthogonal** to the goal of **selecting a pre-trained model without having knowledge of the meta-data used to train the model in the first hand.**
> *   Taskonomy (Zamir et. al.):
>     *   **We can use any model**, whereas Taskonomy is restricted to models trained on the same input with different labels.
>     *   **We examine fine-tuning**, whereas for both methods the encoder part which is transferred is not fine-tuned.
>     *   While Task2Vec and Model2Vec are directly applicable by ranking models based on their similarity in the embedding space, it's unclear how to apply the model search on new tasks using Taskonomy without semantic relations between the upstream and downstream tasks and without training a new network from scratch.
>
> _[The approach is not particularly novel. There is also no novel technical insight that explains the results.]_
>
> We respectfully disagree with the lack of technical insights gained in this work:
>
>
>
> *   The failure of task-aware and task-agnostic methods seems to be non-overlapping, depending on the model pool inspected. This is seen through the success of our hybrid approach, a fact that is very surprising
> *   **The hybrid approach is universal**, e.g., Task2Vec would also benefit from including it
>
> _[The use of the JFT dataset hampers reproducibility since the dataset is not public. I'd like to see results with JFT excluded.]_
>
> The pool ImageNetAccuracies in the appendix does not contain any proprietary model, and it confirms the improvement of the hybrid strategy over the linear proxy task.

---

> > ### Comment · AnonReviewer4 · 2020-11-23
> > **Response to rebuttal**
> >
> > Thanks for your replies!
> >
> > From your reply, I think this paper would be much stronger if you include empirical comparisons to Task2Vec, or improve Task2Vec. I am increasing my rating to borderline accept, but to be honest, the current version would still be a weak paper without such comparisons.

---

### Official Review · AnonReviewer3 · 2020-10-27
**Unclear findings**

**Rating:** 4
**Confidence:** 4

**Review:**

[Summary] This paper presents a large-scale study on model-selection strategies for transfer learning, by performing task-agnostic and task-aware strategies on a large number of models evaluated on a diverse range of tasks.

[Strength] The problem setting is novel and interesting. The proposed quantitative measurement of the quality of selected models, named "regret" is well designed.

[Weakness] The major weakness of this paper is that it seems there is no consistent strategy to out-perform all other methods in every task. Intuitively, task-aware strategies should be better than task-agnostic strategies, but they perform similarly (almost equally) in all model pools, which is quite surprising.

Even if for the advanced strategy hybrid proposed in the latter part of this paper, the optimal pick for this hybrid method is almost identical to the linear evaluation in task-aware strategy in ResNet-50 and expert model pools.

So my biggest concern for this paper is that we don't have a take-home message, other than showing the "No-Free Lunch Theorem" in model selection. So, I hope the authors could re-emphasize what we really learn from this large-scale study.

An interesting direction might be, how we can design a really fast approximation of fine-tuning, so that we can evaluate a model's fitness only by a few iterations (within a very short amount of training time), instead of performing full fine-tuning on the target task.

Considering this limited effective information from this paper, I think it's not suitable for publishing.

---

> ### Author Response · Authors · 2020-11-19
> **Response to AnonReviewer3**
>
> We thank the reviewer for the thoughtful comments.
>
> _[The major weakness of this paper is that it seems there is no consistent strategy to out-perform all other methods in every task.]_
>
> We respectfully disagree: Hybrid strategy is **consistent across all pools** as seen in Figure 6 in the revised draft (previously Figure 5) and Figure 17 in the revised supplementary (previously Figure 16).
>
> _[Even if for the advanced strategy hybrid proposed in the latter part of this paper, the optimal pick for this hybrid method is almost identical to the linear evaluation in task-aware strategy in ResNet-50 and expert model pools.]_
>
> This is correct and expected, since a hybrid strategy with B models receives B-1 top models from linear evaluation. The challenge is in fact to return the experts when needed (which is something other methods are usually not capable of), whilst not missing the generalist model when they perform best.
>
> _[So my biggest concern for this paper is that we don't have a take-home message, other than showing the "No-Free Lunch Theorem" in model selection.]_
>
> We believe that there are several take-home messages:
>
>
>
> *   This is the **first work** that **formulates** the model-search question with a notion of **regret**, on **real-world scenarios** that are most common in practice for **the end user** - choosing the best model with minimal computational demands
> *   We carefully examine cases in which either Task-agnostic or Task-aware methods fail. It is expected that in general they perform well, however, one would like to understand how strong the failures are. In particular, we observe that **failures do not overlap too much**, which is why we believe that this large-scale study sheds a new light on these failures
> *   Finally, we propose a **simple**, **computationally feasible** search strategy - **hybrid** - that captures these failures under one umbrella, consistently outperforming other methods due to the above mentioned property of failures not overlapping often
>
> _[An interesting direction might be, how we can design a really fast approximation of fine-tuning, so that we can evaluate a model's fitness only by a few iterations (within a very short amount of training time), instead of performing full fine-tuning on the target task.]_
>
> We believe that this is out of the scope of our work since understanding the correct method for early stopping is an interesting and difficult problem on its own. There are several reasons for not including such a study in our paper:
>
>
>
> *   The complexity coming with such an approach which goes way beyond training a linear layer. Both proxy tasks that we analyzed **require only a forward pass** (inference) through the pre-trained networks in order to get the representations
> *   Training a large network (or fine-tuning it) robustly with early stopping for mischosen hyperparameters or initialization parameters is an open research problem
> *   Fine-tuning requires additional knowledge about the architecture (e.g., knowing when the batch norm layers are used) of the networks, which a typical user would not be able to grasp for all the pre-trained embeddings. For our proxy task, **no such knowledge is required** until fine-tuning the winning models

---

### Official Review · AnonReviewer1 · 2020-11-02
**Reject due to limited novelty and lack of convincing experiments**

**Rating:** 4
**Confidence:** 5

**Review:**

This paper presents a large scale empirical study on pretrained model selection for transfer learning and show that a hybrid approach that combines task-agnostic and task-aware methods outperforms the existing approaches on VTAB benchmark. The paper is well written and easy to follow. Experiments using 46 pretrained models and 19 downstream tasks show the effectiveness of the hybrid strategy in selecting the right model for transfer learning with low computational complexity.

Overall, I vote for rejecting the paper as the paper has very limited novelty and experiments are not convincing. In particular, I fail to find the major contributions of the paper except the empirical study on VTAB dataset. While papers related to empirical study are interesting and worth of acceptance, this paper does not provide any major insight that could be useful for the future research on transfer learning. Furthermore, many experiments and comparisons are missing which should be included in the paper for a better understanding of the empirical study.

How is the proposed hybrid ranking strategy comparable to the model selection approaches presented in Duality Diagram Similarity: a generic framework for initialization selection in task transfer learning, ECCV 2020; DEPARA: Deep Attribution Graph for Deep Knowledge Transferability, CVPR 2020. These papers should be clearly discussed with possible comparisons in the experiments to show the advantage of the hybrid approach.

Comparison with many simple baselines are missing in the paper. E.g., How does the hybrid strategy comparable to fine-tuning with early stopping. Can we select pre-trained models by finetuning for only few epochs? How does the number of epoch affect the final performance while comparing to the hybrid strategy?

How is the current method comparable to Leep: A new measure to evaluate transferability of learned representations? Experiments and analysis should be included in the experiments to verify the effectiveness of the hybrid strategy.

How does the size of representation/feature affect the final performance? Does the conclusion still hold with different size of features? How does amount of data in the target task affects the performance of ranking? What is the effect of number of pretrained models on the ranking?

Mutual information between the features and discrete labels of the downstream task can be used to rank different models for transfer learning. How does the proposed hybrid strategy related to mutual information based ranking strategy? Experimental comparison should be included in the paper to verify this.

Does the ranking strategy and analysis presented in the paper limited to only classification models? In particular, can models trained using self supervised learning where there is no classification head, be used as pre-trained models in the current approach? How does the analysis change while considering self-supervised models which are now-a-days quite popular in representation learning? What about considering discriminators of generative models, e.g., VAE or BigGAN in the transfer learning analysis? More experiments and analysis should be performed in the experiments.

---

> ### Author Response · Authors · 2020-11-19
> **Response to AnonReviewer1**
>
> We thank the reviewer for the thoughtful comments and pointing out the potentially relevant related work. We have updated the main body of the paper, in particular the "Background" section and the "Other related work" section to improve the positioning of our work, and added Figure 2 for further clarification. That being said, we believe that our contributions are relevant to the research community and the practitioners.
>
> [Contributions]
> **Hybrid approach** is not the main and only contribution, but a byproduct of a careful large scale analysis of current available model search strategies.
>
> [Not limited to classification]
> The proposed method is **not limited to models pre-trained with classification heads, even though we focus only on classification tasks.** Note that VAE and BIGGAN-based models are included (Table 3 of the Appendix).
>
> [Related methods]
> We thank the reviewer for helping us place the work into the more broad research context! While these methods are indeed related in the general sense, they are **not directly comparable to our method:**
>
> Duality diagram similarity (DDS) and DEPARA:
>  - **We can use any model**, whereas Taskonomy is restricted to models trained on the same input with different labels.
>  - **We examine fine-tuning**, whereas for both methods the encoder part which is transferred is not fine-tuned.
>  - The methods aim at computing the results of the Taskonomy dataset faster by only working on a subset of the dataset (the same data for all models).
>  - Adapting DEPARA and DDS to learn some dependency between tasks or models similarly to Task2Vec or Model2Vec would make it a “meta-learned task-aware” strategy, on which we elaborate in the revisited paper.
>  - While Task2Vec and Model2Vec are directly applicable by ranking models based on their similarity in the embedding space, it's unclear how to apply the model search on new tasks using DDS or DEPARA (or taskonomy in general) without semantic relations between the upstream and downstream tasks and without training a new network from scratch.
>
> LEEP:
>  - By including the VTAB models, **we have access to models pre-trained with different loss functions, fully unsupervised to semi-supervised and strongly supervised**, whilst LEEP is only applicable to models pre-trained for classification tasks
>  - LEEP provides theoretical guarantees only for fixed features, even though fine-tuning outperforms the linear classifier in their work.
>  - Linear correlation is not necessarily transitive, which is why LEEP cannot be used directly to derive the relationship between fine-tuning and linear classifier. Hence, we **go one step further than LEEP** in understanding this behaviour.
>
> [Other suggestions]
> Early stopping:
>  - Training a network robustly with early stopping is an open research problem
>  - Fine-tuning requires additional knowledge about the architecture (e.g., knowing when the batch norm layers are used). For our proxy task, **no such knowledge is required**.
>
> Impact of the dimension:
>  - Already shown in Appendix for Pool **_DIM2048_**. The results for this model pool are **consistent** with the pool **_RESNET-50_** in the main body of the paper and were hence relegated to supplementary material.
>
> Mutual Information (MI) based approaches:
>  - MI could be useful, but there is no clear understanding on how to correctly estimate the MI and its relation to transferability.
>
> We are hopeful that, together with the thoroughly revised Section 2, we addressed the major concerns.

---

> > ### Comment · AnonReviewer1 · 2020-11-25
> > **Response after Rebuttal**
> >
> > I thank the authors for their response. However, I am not at all satisfied with the response and hence vote for rejecting the paper because of limited novelty and unconvincing experiments. Below are some of the changes /experiments that I think must be included in the paper before acceptance.
> >
> > - LEEP can be applicable to any classification models whether it is fully supervised or semi-supervised. To the best of my knowledge, It only requires classification head. So, why it can not be applied to VTAB models is not clear to me.
> > - I agree that early stopping is an open research problem. However, it can be used as a simple baseline in the comparison, e.g., finetuning with only 1 epoch or finetuning with only 10 epochs. I don't agree with the authors that finetuning requires when the batch norm layers are being used. Why can't I finetune a model for 5 epoch without knowing anything about BN layers and then use the premature models’ test accuracies as ranking scores.
> > - Also, why can't we simply rank the checkpoints by their mutual information between the high-dimensional features and discrete labels of the downstream task? It is a variational lower bound parameterized by a neural network. See: "Belghazi et al. Mine: mutual information neural estimation. arXiv preprint arXiv:1801.04062, 2018".
> > - Moreover, How does amount of data in the target task affects the performance of ranking? What is the effect of number of pretrained models on the ranking?
> >
> > Based on the above missing experiments, I am keeping my score same as the initial one.

---

> > > ### Author Response · Authors · 2020-11-25
> > > **Discussion**
> > >
> > > We thank the reviewer for his comments. However, we disagree with the proposed suggestions as they are either orthogonal to our method, or have been shown as inefficient, by other related work.
> > >
> > >
> > >
> > > *   LEEP: This method assumes a pre-trained classification head as part of the pre-trained model, which works for fully/semi-supervised. However, some models from the VTAB benchmark do not fulfill this requirement! (e.g., feature representations originating from a GAN or VAE which are part of the models in the pool)
> > > *   Batch Norm requires one to adjust the parameters to the new data distribution. Reporting the behaviour of using restricted finetune compute to estimate unrestricted finetune compute accuracy is possible, however that is clearly as (if not more) sensitive of hyperparameter selection. Instead, we focused on doing an extensive study of methods which are known to be less sensitive to hyperparameter search to identify the best models, which are then used with larger finetune compute.
> > > *   Whilst one could use “MINE”, it is known that estimating the MI suffers considerably from a bias - variance tradeoff and no method bypasses those issues (see “Poole, B., Ozair, S., Van Den Oord, A., Alemi, A., & Tucker, G. (2019, May). On Variational Bounds of Mutual Information. In ICML.”). The suggested approach “MINE” falls into this category and additionally has a large computational requirement as it requires to train a neural network which makes it unusable as a “cheap” proxy task.
> > > *   We restricted ourselves to the 1K samples setting from VTAB on purpose, as it is known that this yields the largest gain in using transfer learning compared to training from scratch. The impact of the size of the model pool is directly visible via the novel definition of regret which we consider as a major contribution compared to other performance proxies.

---

### Decision · Program_Chairs · 2021-01-07
**Final Decision**

**Decision:**

Reject

**Comment:**

The paper studies efficient strategies for selection of pre-trained models for a downstream task. The main concerns consistently raised by the reviewers were limited methodological novelty, insufficient experimental analysis, unclear findings, and positioning of the paper with respect to related work that was ignored in the initial version. After the author response, R4 raised the score to borderline accept (still indicating the paper is weak without proper comparisons with other methods), whereas all other reviewers remained negative. The paper does have merits, as the methods are simple, and the problem is very practical (and somewhat understudied). However, the AC agrees with the majority that the paper is not ready for ICLR. The novelty is limited and the paper would benefit from more experiments, such as comparisons with simple baselines like early stopping as indicated by R1 and R3, and other methods such as Task2vec which address the same problem. The authors are encouraged to revise the paper according to the reviewers comments and submit it to another top conference.